# Proximate ferromagnetic state in the Kitaev model material $\alpha$-RuCl$_3$

H. Suzuki [1✉], H. Liu [1✉], J. Bertinshaw[1], K. Ueda[1,10], H. Kim[1,2,3], S. Laha [1], D. Weber [1,11], Z. Yang[1], L. Wang[1], H. Takahashi [1], K. Fürsich [1], M. Minola [1], B. V. Lotsch [1,4], B. J. Kim [1,2,3], H. Yavaş[5,12], M. Daghofer[6,7], J. Chaloupka[8,9], G. Khaliullin [1], H. Gretarsson[1,5] & B. Keimer [1✉]

$\alpha$-RuCl$_3$ is a major candidate for the realization of the Kitaev quantum spin liquid, but its zigzag antiferromagnetic order at low temperatures indicates deviations from the Kitaev model. We have quantified the spin Hamiltonian of $\alpha$-RuCl$_3$ by a resonant inelastic x-ray scattering study at the Ru $L_3$ absorption edge. In the paramagnetic state, the quasi-elastic intensity of magnetic excitations has a broad maximum around the zone center without any local maxima at the zigzag magnetic Bragg wavevectors. This finding implies that the zigzag order is fragile and readily destabilized by competing ferromagnetic correlations. The classical ground state of the experimentally determined Hamiltonian is actually ferromagnetic. The zigzag state is stabilized by quantum fluctuations, leaving ferromagnetism – along with the Kitaev spin liquid – as energetically proximate metastable states. The three closely competing states and their collective excitations hold the key to the theoretical understanding of the unusual properties of $\alpha$-RuCl$_3$ in magnetic fields.

[1] Max-Planck-Institut für Festkörperforschung, Stuttgart, Germany. [2] Department of Physics, Pohang University of Science and Technology, Pohang, South Korea. [3] Center for Artificial Low Dimensional Electronic Systems, Institute for Basic Science (IBS), Pohang, South Korea. [4] Department of Chemistry, University of Munich (LMU), München, Germany. [5] Deutsches Elektronen-Synchrotron DESY, Hamburg, Germany. [6] Institute for Functional Matter and Quantum Technologies, University of Stuttgart, Stuttgart, Germany. [7] Center for Integrated Quantum Science and Technology, University of Stuttgart, Stuttgart, Germany. [8] Department of Condensed Matter Physics, Faculty of Science, Masaryk University, Brno, Czech Republic. [9] Central European Institute of Technology, Masaryk University, Brno, Czech Republic. [10] Present address: Department of Applied Physics, University of Tokyo, Tokyo, Japan. [11] Present address: Institute of Nanotechnology, Karlsruhe Institute of Technology, Karlsruhe, Germany. [12] Present address: SLAC National Accelerator Laboratory, Menlo Park, CA, USA. ✉email: H.Suzuki@fkf.mpg.de; H.Liu@fkf.mpg.de; B.Keimer@fkf.mpg.de

Quantum spin liquid states are characterized by a large degree of entanglement that supports fractionalized quasiparticles[1,2]. The Kitaev model on a honeycomb lattice[3] has been a central focus of research in recent years, due to its exact solubility and its quantum spin-liquid ground state. Notably, the elementary excitations in the presence of a magnetic field are represented by emergent non-abelian anyons, which could serve as a key element in topological quantum computation. The bond-directional magnetic interactions of the Kitaev model can be realized in strongly correlated transition metal compounds, where spin–orbit entangled pseudospins $\widetilde{S} = 1/2$ are arranged on a honeycomb lattice of edge-shared octahedra[4,5]. Honeycomb-lattice compounds composed of $Ir^{4+}$ or $Ru^{3+}$ ions are prime candidates for the experimental realization of a Kitaev spin liquid[6–8], because their $t_{2g}^5$ electron configuration supports $\widetilde{S} = 1/2$ states in the presence of strong spin–orbit coupling[9]. In particular, $\alpha$-RuCl$_3$ (hereafter RuCl$_3$)[10], a prototypical example of two-dimensional van-der-Waals magnetism[11], has been the focus of intensive research thanks to the availability of large single crystals and perspectives for the synthesis of functional devices from exfoliated nanosheets[12].

Most of the Kitaev candidate materials, however, undergo magnetic transitions at sufficiently low temperatures. This is principally caused by non-Kitaev nearest neighbor (NN) interactions including Heisenberg and off-diagonal couplings[13–15] that originate from direct hopping between the $\widetilde{S} = 1/2$ ions and from the distortions of their coordination octahedra. The additional interactions in real materials call for analysis of the extended Kitaev–Heisenberg Hamiltonian, whose theoretical phase diagram in parameter space is dominated by a variety of magnetically ordered phases. In particular, the zigzag antiferromagnetic (AFM) state, which is realized in RuCl$_3$[16], is predicted in a wide parameter range adjacent to the pure Kitaev points[14,15,17]. Longer-range Heisenberg interactions tend to further stabilize the zigzag order[18–21], driving the system away from the spin-liquid phase.

Despite the magnetic ordering of real materials at low temperatures, the Kitaev interactions can manifest themselves in the dynamical spin correlations. In the pure Kitaev model, signatures of emergent quasiparticles appear in the form of an excitation continuum in the spin dynamical structure factor[22–24]. In RuCl$_3$,

a magnetic scattering continuum has been observed by Raman scattering[25] and inelastic neutron scattering experiments[26–29]. Furthermore, the observation of half-integer quantization of the thermal Hall transport coefficient in a magnetic field[30] supports fractionalization of the spins into Majorana fermions. With increasing experimental evidence for Kitaev interactions in RuCl$_3$, it is crucial to fully determine its pseudospin Hamiltonian. A set of parameters that coherently accounts for the low-temperature zigzag order and the signatures of fractionalization would establish a concrete, controllable pathway to the spin-liquid phase. This objective has, however, not yet been achieved, partly because different interaction terms have been emphasized to explain limited sets of experimental data, leading to a zoo of proposed pseudospin models[31–33].

In the present work, we have determined the leading terms in the pseudospin Hamiltonian of RuCl$_3$ by using resonant inelastic x-ray scattering (RIXS)[34,35] at the Ru $L_3$ absorption edge to investigate the excitation spectra of RuCl$_3$ over a wide spectral range. The resonant enhancement of the cross section enables the observation of high-energy excitations (>100 meV) with high statistics, allowing us to accurately determine the cubic crystal field splitting $10Dq$, the Hund's coupling $J_H$, and the spin–orbit coupling constant $\lambda$; these parameters were then used for a theoretical evaluation of the exchange constants. By utilizing the large momentum transfer between the incoming and outgoing photons at the Ru $L_3$ absorption edge (2837.8 eV), we mapped out the intensity of magnetic excitations in the low-energy $\widetilde{S} = 1/2$ manifold across the entire first Brillouin zone. Comparison with exact-diagonalization calculations of the RIXS intensity yields the hierarchy of interaction parameters between the pseudospins and reveals their individual roles: (1) the ferromagnetic (FM) Kitaev coupling $K = -5.0$ meV is dominant, (2) the FM Heisenberg interaction $J = -3.0$ meV enhances the FM correlations and renders the zigzag order fragile, (3) the off-diagonal interaction $\Gamma = 2.5$ meV stabilizes the zigzag order at low temperatures and explains the magnetic moment direction. In fact, our experiments demonstrate that there is a strong competition between the zigzag and ferromagnetic states, which is resolved in favor of the former by quantum zero-point fluctuations, as illustrated in Fig. 1a. The interaction Hamiltonian obtained in this way is in good agreement with the one obtained from our theoretical analysis of the high-energy multiplets, and thus provides a solid foundation for

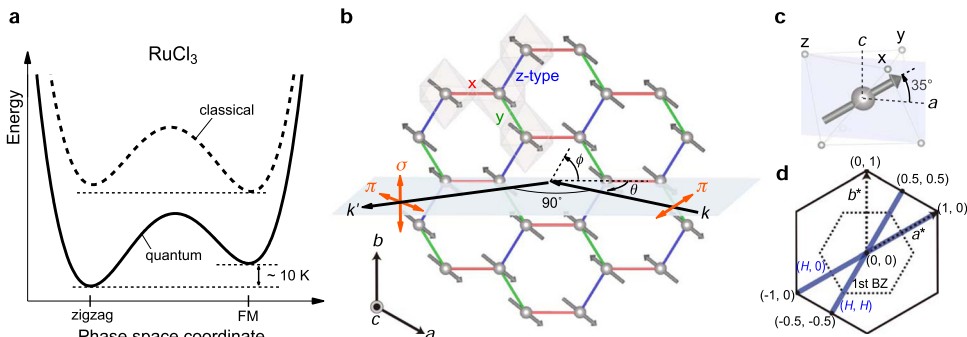

**Fig. 1 Phase competition in RuCl$_3$ revealed by Ru $L_3$ edge RIXS, and the scattering geometry. a** Schematic of classical and quantum energy landscapes in the vicinity of the zigzag ground state. The zigzag order is stabilized by quantum effects, and only slightly lower in energy than the competing metastable ferromagnetic (FM) state. **b** The pseudospin-1/2 moments (gray arrows) show the zigzag-type magnetic order pattern on the honeycomb lattice of $\alpha$-RuCl$_3$. The gray-shaded RuCl$_6$ octahedra share the edges on the three distinct $x$, $y$, and $z$-type bonds, represented by the red, green, and blue lines, respectively. $a$, $b$, and $c$ represent the crystallographic lattice vectors. The incident x-ray photons with momentum $\boldsymbol{k}$ are linearly $\pi$-polarized and the polarization of the scattered photons with momentum $\boldsymbol{k}'$ is not analyzed. The scattering angle is fixed at 90° and the in-plane momentum transfer $\boldsymbol{q}$ is changed by rotating the sample angle $\theta$. The azimuthal angle $\phi$ is used to change the measurement path. **c** Local moment direction in the RuCl$_6$ octahedron. The directions from the central Ru atom to the adjacent Cl atoms denoted by $x$, $y$, and $z$ define the local $x$, $y$, and $z$ coordinate axes. The magnetic moment lies within the $ac$ plane and points 35° from the $a$ axis[68,53]. **d** $\boldsymbol{q} = (H, 0)$ and $(H, H)$ paths investigated in the RIXS experiment. The dotted hexagon indicates the first Brillouin zone (BZ).

further work on RuCl$_3$, including the theoretical analysis of the purported spin-liquid behavior in magnetic fields. More generally, our comprehensive approach to the determination of the low-energy effective Hamiltonian can serve as a blueprint for research on other quantum magnets and spin-liquid candidates.

## Results

**Crystal structure and scattering geometry**. Figure 1b shows the crystal structure and zigzag magnetic ordering of RuCl$_3$, as well as the scattering geometry for the RIXS experiment. To facilitate comparison with theoretical analysis, we will use the hexagonal crystallographic notation with $a = b = 5.96$ Å and $c = 17.2$ Å, where the $ab$ plane corresponds to the honeycomb plane and the $c$ axis is perpendicular to it. The distinct $x$, $y$, $z$ type bonds are represented by red, green, and blue lines, respectively. The incident x-ray photons were $\pi$-polarized and the polarization of the scattered photons was not analyzed. Hereafter, the momentum transfer is expressed in terms of the in-plane component $\boldsymbol{q}$, which was scanned by rotating the sample angle $\theta$. Figure 1c shows the definition of the local $xyz$ coordinates and the local moment direction within the RuCl$_6$ octahedron. In the following theoretical analysis, the parameters in the pseudospin Hamiltonian are chosen to reproduce the moment direction. Figure 1d shows the measurement paths in the $\boldsymbol{q}$-space investigated in our RIXS experiment. We performed the measurements along the $\boldsymbol{q} = (H, 0)$ and $(H, H)$ directions, by setting the azimuthal angle $\phi$ to 0° and −30°, respectively. $\boldsymbol{q}$ is expressed in reciprocal lattice units (r.l.u.). Unless otherwise stated, the measurements were performed at the base temperature of 20 K, in the paramagnetic state.

**High-energy multiplets**. Figure 2 provides an overview of the RIXS spectrum at the Brillouin zone center [$\boldsymbol{q} = (0, 0)$] over a wide range of excitation energies. A broad continuum emerging above the charge gap of ~1 eV (dashed blue line in Fig. 2) and extending up to at least 4 eV can be assigned to intersite electron-hole excitations, consistent with the continuum observed by optical spectroscopy[36] and electron energy loss spectroscopy[37]. On top of the intersite continuum, one observes the main peak B and the shoulder structures $\alpha$, $\beta$, and $\gamma$, which are assigned to intra-ionic crystal-field transitions from the $t_{2g}^5$ ground state to Hund's multiplets within the $t_{2g}^4 e_g^1$ manifold. Below the charge gap (<1 eV), a pronounced peak (A$_1$) appears at 0.25 eV, which

originates from transitions from the ground state $\widetilde{S} = 1/2$ doublet to the excited $\widetilde{S} = 3/2$ quartet. This phenomenology establishes the notion of a low-energy $\widetilde{S} = 1/2$ doublet constituting the pseudospin Hamiltonian. We ascribe the small shoulder structure A$_2$ to multiples of the A$_1$ exciton. The spectral lineshape resembles that of Ru $M$-edge RIXS data[38], but the better statistics of the present data allows the precise determination of the microscopic parameters from the multiplet analysis (Supplementary Note 2).

The theoretical RIXS intensity with the optimal parameters $10Dq = 2.4$ eV, $J_{\rm H} = 0.34$ eV, and $\lambda = 0.15$ eV is shown as vertical bars in Fig. 2. These parameters are in good agreement with previous reports[10,36,38,39] and will be used below to quantify the exchange constants. The theoretical result clearly captures the peak energies and intensities of the crystal field multiplets (B, $\alpha$, $\beta$, and $\gamma$) that are located around $10Dq$ and split as a function of $J_{\rm H}$, and the $\widetilde{S} = 3/2$ transitions (A$_1$) located at ~$3\lambda/2$. Note that there is no discernible splitting of the A$_1$ peak. This is in contrast to the clear trigonal crystal field splitting observed in the honeycomb iridates A$_2$IrO$_3$ (A = Na, Li)[40], and indicates that the trigonal field splitting in RuCl$_3$ is smaller than the experimental energy resolution of ~0.1 eV. Indeed, from an analysis of the magnetic susceptibility anisotropy, we obtain a $\widetilde{S} = 3/2$ quartet splitting of only $\simeq 30$ meV (Supplementary Note 4).

**Spin−orbit excitons and quasi-elastic peak**. Figure 3 shows the momentum dependence of the raw RIXS spectra along the $\boldsymbol{q} = (H, 0)$ and $(H, H)$ directions in the low-energy range. The overall monotonic decrease of the intensity from $H < 0$ (grazing incidence) to $H > 0$ (grazing exit) is due to the geometrical effect of x-ray self-absorption[41], which will be accounted for in the following quantitative intensity analysis. The A$_1$ peak shows no energy dispersion along the $(H, 0)$ and $(H, H)$ directions, demonstrating the nearly localized nature of the $\widetilde{S} = 3/2$ excitons. Notably, at $T = 20$ K the quasi-elastic peak intensity does not show any local maximum at the zigzag magnetic Bragg wavevectors $\boldsymbol{q} = (\pm 0.5, 0)$, indicating that short-range zigzag correlations are quickly suppressed when magnetic long-range order disappears at $T_{\rm N} = 7$ K. This observation is in sharp contrast to the robust zigzag correlations that persist far above $T_{\rm N}$ in Na$_2$IrO$_3$[42,43]. This finding suggests that the energy landscape of RuCl$_3$ has only shallow minima around the zigzag ordered states (Fig. 1a), and that closely competing states with characteristic vectors $\boldsymbol{q} \sim 0$, such as the ferromagnetic one, exist as metastable states with energies of the

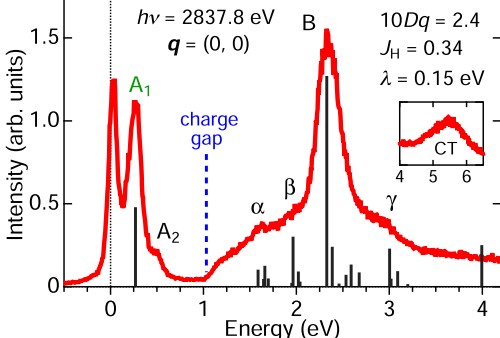

**Fig. 2 Determination of microscopic parameters from high-energy multiplets.** A representative Ru $L_3$ RIXS spectrum of RuCl$_3$ at $\boldsymbol{q} = (0, 0)$, taken with photons of energy 2837.8 eV. The dashed blue line indicates the onset energy of the intersite electron-hole continuum. The vertical black bars indicate the theoretical RIXS intensity from ionic multiplet calculations (Supplementary Note 2). The cubic crystal field splitting ($10Dq$), the Hund's coupling ($J_{\rm H}$), and the spin-orbit coupling ($\lambda$) parameters obtained from an analysis of the RIXS spectrum are indicated. The inset shows the high-energy region comprising the $p$-$d$ charge-transfer (CT) excitations.

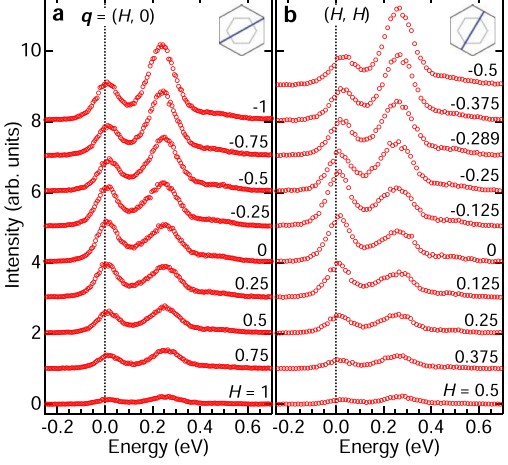

**Fig. 3 Momentum dependence of low-energy RIXS spectra. a, b** Low-energy Ru $L_3$ RIXS spectra at $T = 20$ K along the $\boldsymbol{q} = (H, 0)$ and $(H, H)$ directions. The insets show the measurement paths in the $\boldsymbol{q}$ space.

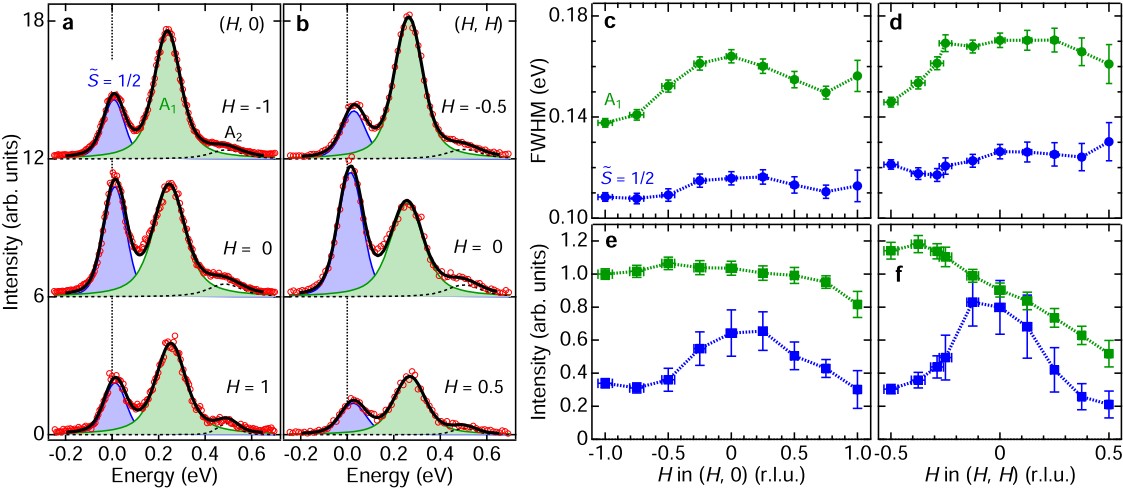

**Fig. 4 FWHM and intensity of the A$_1$ peak and $\widetilde{S} = 1/2$ excitations. a, b** Ru $L_3$ RIXS spectra after the correction of x-ray self-absorption effect. The correction to the raw spectra was performed following the procedure in ref. [41]. The blue, green, and dashed black curves exemplify the spectral decomposition into three Voigt profiles representing the quasi-elastic peak, the $\widetilde{S} = 3/2$ transitions (A$_1$), and the multi-excitons (A$_2$), respectively. The total fitted curves are shown as thick black lines. **c, d** The full-width at half maximum (FWHM) of the quasi-elastic and the A$_1$ peaks as functions of $\boldsymbol{q}$. **e, f** Momentum dependence of peak intensities along the $\boldsymbol{q} = (H, 0)$ and $(H, H)$ directions taken at $T = 20$ K.

order of $k_B T_N \sim 1$ meV. This energy scale is roughly consistent with the Zeeman energy of $\widetilde{S} = 1/2$ under a magnetic field of ~8 T, where the zigzag order disappears[44] and signatures of a field-induced quantum spin-liquid emerge[29,45,46].

To perform a quantitative analysis of the RIXS spectra, we have corrected the raw RIXS intensity for the effect of x-ray self-absorption, following the procedure described in Minola et al.[41] (see Supplementary Note 5 for details). Figure 4a, b shows representative corrected spectra along the $\boldsymbol{q} = (H, 0)$ and $(H, H)$ directions. We decomposed these spectra into three Voigt profiles, representing quasi-elastic scattering (blue), $\widetilde{S} = 3/2$ excitons (green), and multi-excitons (dashed black). Here, the Gaussian full-width at half maximum (FWHM) of the Voigt profiles was fixed at the energy resolution of 0.1 eV.

We highlight two characteristic observations that are apparent in the decomposed spectra. First, the A$_1$ peak is significantly broader than the quasielastic peaks. It is tempting to associate the extra broadening directly with the splitting of the $\widetilde{S} = 3/2$ transitions due to the trigonal field $\Delta$, as suggested in Lebert et al.[38] based on Ru $M$-edge RIXS data. However, Ru $M$-edge RIXS is incapable of fully quantifying the $\boldsymbol{q}$ dependence due to the small momentum transfer of Ru $M$-edge x-rays (461 eV). The full momentum dependence of the FWHM extracted from our $L$-edge data requires careful reconsideration of this issue. Figure 4c, d shows that the FWHM of the quasi-elastic peak is almost $\boldsymbol{q}$-independent and only slightly larger than the energy resolution (0.1 eV), reflecting the relatively small bandwidth of magnetic excitations of the $\widetilde{S} = 1/2$ sector. On the other hand, the A$_1$ FWHM is ~50 meV wider than the quasi-elastic peak and has a broad maximum at the Γ point both along the $(H, 0)$ and $(H, H)$ directions. Since the $\widetilde{S} = 3/2$ states have orbital degeneracy and are Jahn-Teller active[47], the orbital interactions and coupling to phonons are possible reasons of the overall broadening. Note that the maximum of the FWHM occurs concomitantly with a slight deviation of the fitting curves from the data points in the low-energy tail of the A$_1$ peak [see $H = 0$ curves in Fig. 4a, b], whereas at large $|H|$ the lineshape is perfectly captured by the Voigt profiles. This suggests that the A$_1$ peak contains not only the intra-ionic transitions but also an additional feature around the Γ point at lower energy. Here we refer to the case of Na$_2$IrO$_3$, where

an excitonic bound state was observed below the $\widetilde{S} = 3/2$ transition around the Γ point[40]. We expect that the same phenomenology also applies to RuCl$_3$ with a reduced energy scale. An important lesson here is that the width of the A$_1$ peak is momentum dependent, and hence cannot be directly linked to the splitting of the $\widetilde{S} = 3/2$ transitions by the trigonal field $\Delta$. In fact, we will determine this splitting based on the $g$-factor anisotropy and find that it is significantly smaller than the additional broadening of the A$_1$ peak (Supplementary Note 4).

Second, one observes a distinct $\boldsymbol{q}$ dependence of the scattering intensities. Figure 4e, f shows the integrated intensity (defined as the total area of the decomposed peak) as a function of $\boldsymbol{q}$. While the A$_1$ peak shows a monotonic intensity decrease as $H$ increases, the quasielastic peak exhibits broad intensity maxima around $\boldsymbol{q} = (0, 0)$ both along the $(H, 0)$ and $(H, H)$ directions. We note that the quasi-elastic intensity from extrinsic scattering such as thermal diffuse scattering, surface roughness, and from the tail of the specular reflectivity is negligible (see Supplementary Note 1). In the present 90° scattering geometry with $\pi$-polarized incident x-rays, the charge (Thomson) scattering is strongly suppressed because the polarization of the incident x-ray photons is always perpendicular to the one of the outgoing photons[35]. Indeed, the maxima of the quasi-elastic peaks shown in Fig. 4a, b are located at positive energy, demonstrating that intrinsic magnetic excitations dominate the spectral weight. One further notices that the $\boldsymbol{q}$ dependence around the Γ point is more sharply peaked along the $(H, H)$ direction than along the $(H, 0)$ direction, in qualitative agreement with the star-shaped excitation continuum observed by inelastic neutron scattering experiments[27,28].

## Theoretical analysis

**Model Hamiltonian and method.** The momentum dependence of the quasi-elastic peaks is a signature of the spatial correlations among the pseudospins, and thus enables one to access the exchange interaction parameters. To describe the pseudospin $\widetilde{S} = 1/2$ excitations and the corresponding RIXS intensity, we employ the extended Kitaev–Heisenberg model $\mathcal{H}_{ij}^{(\gamma)}$, supplemented by the third-NN Heisenberg interaction $J_3 \widetilde{S}_i \cdot \widetilde{S}_j$ which is essential to stabilize the zigzag-type magnetic order. For the $z$-type bonds,

$\mathcal{H}_{ij}^{(z)}$ reads as

$$\mathcal{H}_{ij}^{(z)} = K\widetilde{S}_i^z\widetilde{S}_j^z + J\widetilde{\boldsymbol{S}}_i \cdot \widetilde{\boldsymbol{S}}_j + \Gamma(\widetilde{S}_i^x\widetilde{S}_j^y + \widetilde{S}_i^y\widetilde{S}_j^x) \\ + \Gamma'(\widetilde{S}_i^z\widetilde{S}_j^x + \widetilde{S}_i^x\widetilde{S}_j^z + \widetilde{S}_i^y\widetilde{S}_j^z + \widetilde{S}_i^z\widetilde{S}_j^y). \quad (1)$$

For the $\gamma = x, y$-type bonds, $\mathcal{H}_{ij}^{(\gamma)}$ follow from cyclic permutations of $\widetilde{S}^x$, $\widetilde{S}^y$, and $\widetilde{S}^z$.

Based on the small trigonal field $\Delta$ (see Supplementary Note 4), we assume that $\Gamma'$ is small as prior work has shown that this term is zero at cubic symmetry[14,48]. We take the model parameters of Winter et al.[20], which are widely used in the literature, as a starting point of our analysis and optimize the parameter set to fit the RIXS data, subject to the condition that they reproduce the ordered moment direction as well.

To this end, we first apply the method of spin-coherent states[49] and perform a systematic scan of the moment direction through parameter space (see Fig. 4 of ref. [21] for an illustration). Having identified the relevant areas of parameter space, we simulate the RIXS intensity at nonzero temperatures by utilizing the Thermal Pure Quantum (TPQ) method[50,51] for two hexagon-shaped clusters of 24 and 32 sites, each with a distinct set of accessible $\boldsymbol{q}$ vectors. Specifically, we calculate the equal-time pseudospin correlation function $\langle\widetilde{S}_{\boldsymbol{q}}^\alpha\widetilde{S}_{-\boldsymbol{q}}^\beta\rangle$ $(\alpha, \beta = x, y, z)$ and combine its components according to ref. [52] to construct the integrated RIXS intensity for our scattering geometry. This intensity is averaged over many realizations of the TPQ state to reduce the statistical errors. As can be seen in Fig. 5c, d, finite-size effects are negligible at the temperatures of interest here.

**Momentum dependence of quasi-elastic intensity and exchange constants.** Figure 5a, b shows the $\widetilde{S} = 1/2$ intensity data along the $\boldsymbol{q} = (H, 0)$ and $(H, H)$ directions, respectively, together with the theory curves calculated at several temperatures for the optimal parameter set $K = -5$, $J = -3$, $\Gamma = 2.5$, $\Gamma' = 0.1$, and $J_3 = 0.75$ meV. The theory curves were generated by interpolating the original data points of the 24- and 32-site clusters [see Fig. 5c, d].

The observed $\boldsymbol{q}$ dependence of the RIXS intensity at 20 K is well reproduced by our calculations, including in particular the intensity maximum at $(0, 0)$ and the sharper intensity profile along the $(H, H)$ direction. Note that the theory curve for the $(H, 0)$ direction calculated at 5 K has pronounced peaks at $\boldsymbol{q} = (\pm 0.5, 0)$ corresponding to the zigzag magnetic order that sets in at $T_N \simeq 7$ K. However, these peaks quickly vanish at 20 K leading to the absence of local maxima in the RIXS intensity, while the other regions of the spectra show only gradual modifications.

Having demonstrated the capability of our theoretical approach to the RIXS intensity, we apply our methodology to other models in the literature to test their validity. Figure 5c, d shows a comparison of the theoretical RIXS intensity at 20 K for different parameter sets. Specifically, we compare our model $(K, J, \Gamma, \Gamma', J_3) = (-5, -3, 2.5, 0.1, 0.75)$ with that of Winter et al.[20] $(-5, -0.5, 2.5, 0, 0.5)$ and the model 2 of Sears et al.[53] $(-10, -1.5, 8.8, 0, 0.4)$ meV. In contrast to our data, the two models in the literature show that the global maximum is not located around $(0, 0)$ but stays instead around the magnetic Bragg wavevectors $(\pm 0.5, 0)$. The experimental intensity maximum around the $\Gamma$ point therefore highlights enhanced FM correlations, which we ascribe to the large FM Heisenberg interaction $J = -3$ meV $\sim K/2$ (compared to $J/K = 0.1$[20] and $0.15$[53]). This comparison shows that the momentum dependence of the RIXS intensity is highly sensitive to the model parameters. Although they are subject to certain variations (e.g., by including the interlayer couplings[54,55] in the fits), their overall hierarchy obtained above is robust, as dictated by the condition that the $\boldsymbol{q} \sim 0$ correlations, supported by FM Kitaev and FM Heisenberg couplings, are closely competing with the zigzag order and become prominent already at $T \sim 20$ K. The proximity to the FM is pointed out also in recent theoretical works[33,56], based on the analysis of low-energy magnetic excitations.

We note that the proximity to the FM state has important implications for the theoretical description of RuCl$_3$. Table 1 shows the classical energy of the FM state (with respect to that of the zigzag phase) in several models. In contrast to the models of refs. [20] and [53], the ferromagnetic state in our RIXS-derived model

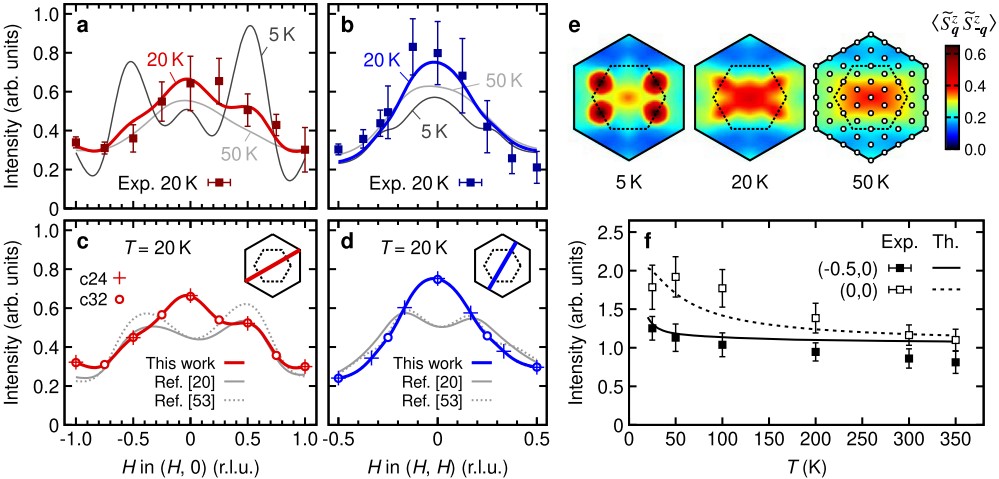

**Fig. 5 Theoretical RIXS intensity and pseudospin correlations in the Kitaev–Heisenberg model. a**, **b** Momentum dependence of the theoretical RIXS intensity calculated at $T = 5$ K (dashed), 20 K (solid blue), and 50 K (solid cyan), using the pseudospin Hamiltonian [Eq. (1)]. To obtain the best fit to the experimental data (squares), the exchange parameters $K = -5$, $J = -3$, $\Gamma = 2.5$, $\Gamma' = 0.1$ and $J_3 = 0.75$ meV were used. **c**, **d** Comparison of the RIXS intensity computed with different parameter sets. The optimal theoretical curve is compared with those for the parameter sets proposed in refs. [20] and [53]. The insets show the $\boldsymbol{q}$ paths. The points represent results at the accessible $\boldsymbol{q}$ vectors for the 24-site cluster (crosses) and 32-site cluster (circles) that were used to construct the smooth profiles. **e** Temperature evolution of the equal-time pseudospin correlation function $\langle\widetilde{S}_{\boldsymbol{q}}^z\widetilde{S}_{-\boldsymbol{q}}^z\rangle$ for the optimal parameter set. The maps were calculated for the 32-site cluster with the accessible $\boldsymbol{q}$ vectors (circles) marked on the 50 K map. **f** Temperature dependence of the RIXS intensity at $\boldsymbol{q} = (0, 0)$ and $(-0.5, 0)$. The data points were collected with the azimuthal angle of $\phi = 0$ [the geometry for the $(H, 0)$ path]. The lines show the theoretical curves obtained by a simulation for the 24-site cluster.

**Table 1 Classical energy of ferromagnetic state with respect to zigzag state for representative model parameters (all energies in meV).**

| Models | $K$ | $J$ | $\Gamma$ | $\Gamma'$ | $J_3$ | $E_{FM}-E_{ZZ}$ |
|---|---|---|---|---|---|---|
| This work | −5 | −3 | 2.5 | 0.1 | 0.75 | −0.12 |
| Winter et al.[20] | −5 | −0.5 | 2.5 | 0 | 0.5 | 0.37 |
| Sears et al.[53] | −10 | −1.5 | 8.8 | 0 | 0.4 | 0.50 |

is actually lower in energy (−0.12 meV) than the zigzag state on the classical level, as illustrated in Fig. 1a. The stabilization of the zigzag order is therefore ascribed to strong quantum fluctuations, which are intrinsic to highly frustrated Kitaev interactions and fully accounted for in our exact-diagonalization analysis. This observation invalidates the classical treatment of the zigzag order in RuCl₃ based on, e.g., the standard linear spin-wave theory. Instead, quantum zero-point fluctuations have to be considered from the outset, in order to obtain a stable zigzag order and to correctly describe its excitations. The quantum origin of zigzag order and the proximity to the FM state should be essential to understand the anomalous neutron scattering continuum in RuCl₃[26–29], and also its field-induced properties, given the nontrivial topology of the ferromagnetic magnons in the Kitaev materials[57,58].

To further visualize the characteristics of our model, we plot in Fig. 5e intensity maps of the equal-time pseudospin correlation function $\langle \tilde{S}^z_q \tilde{S}^z_{-q} \rangle$ for $T = 5$ K, 20 K, and 50 K. The maps were generated from the 32-site cluster with the accessible **q** vectors (circles) marked on the 50 K map. At 5 K, the system is in the zigzag ordered phase, and the spectral weight is concentrated around the magnetic Bragg wavevectors, yet sizeable FM correlations are also evident from the intensity around (0, 0). As the zigzag long-range order disappears, the spectral weight at the Bragg wavevectors is quickly transferred to the vicinity of (0, 0), reflecting pronounced FM correlations due to the FM Kitaev and Heisenberg interactions [$T = 20$ K Fig. 5e]. At $T = 50$ K, the intensity profile is fully dominated by the Kitaev term and resembles that of the pure FM Kitaev model. To further demonstrate the predictive power of our model, we show in Fig. 5f the temperature dependence of the RIXS intensity at **q** = (0, 0) and (−0.5, 0). The data were collected with the azimuthal angle of $\phi = 0$ [i.e., the geometry for the (H, 0) path]. The RIXS intensities at **q** = (0, 0) and (−0.5, 0) show a gradual decrease up to ~200 K and converge at higher temperatures, in agreement with the theory. Notably, the **q** dependence is clearly present even at $T = 100$ K, manifesting the strong correlations among pseudospins well above $T_N$.

**Theoretical estimation of K, J, Γ, and Γ′.** The dominance of the FM Kitaev coupling found experimentally is consistent with theoretical considerations[4] of the impact of the Hund's-rule coupling $J_H$ on the exchange interactions. The sizeable value of Γ can be attributed to the fact that the 4d orbitals are spatially extended so that their direct overlap $t'$ is large[14]. The relatively large ($J \sim K/2$) Heisenberg coupling of FM sign is somewhat surprising. It might be supported by interorbital $t_{2g}$-$e_g$ hoppings[5], given that the cubic splitting $10Dq$ is rather small here.

Having at hand several microscopic parameters such as $J_H$, $10Dq$, $\Delta_{pd}$, and $\lambda$ quantified by the RIXS data above, we can estimate the exchange parameters from theory. In particular, we would like to evaluate the non-diagonal Γ′ term (neglected in the fits above) as a function of the trigonal crystal field $\Delta$, and see if it is indeed small at $\Delta$ values realistic for RuCl₃. To this end, we follow previous work[17,59] and consider four different NN

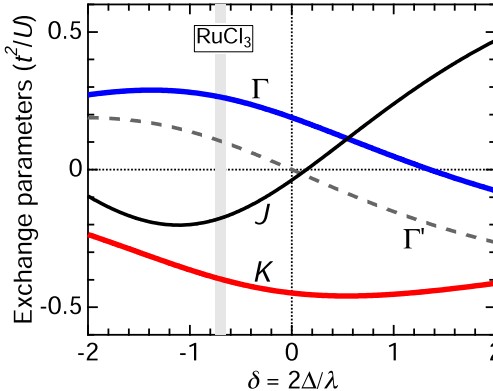

**Fig. 6 Theoretical exchange parameters.** Exchange parameters $K$ (red), $J$ (black), Γ (blue), and Γ′ (gray) as a function of $\delta = 2\Delta/\lambda$, calculated using $10Dq = 2.4$ eV, $J_H = 0.34$ eV, and the $pd$ charge-transfer gap $\Delta_{pd} = 5.5$ eV as obtained from the RIXS data (Fig. 2). We use representative values of the Coulomb interaction $U = 2.5$ eV for the Ru-$d$ orbitals, $U_p = 1.5$ eV and Hund's coupling $J_p = 0.7$ eV for the Cl-$p$ orbitals, a ratio of $t_{pd\sigma}/t_{pd\pi} = 2$ between the $pd$ charge-transfer integrals in the $\sigma$ and $\pi$ channels, and a direct $t_{2g}$ hopping $t' = 0.4t$. The exchange constants are given in units of $t^2/U$. The vertical gray line indicates $\delta \approx -0.7$ appropriate for RuCl₃.

exchange mechanisms: (1) indirect hopping $t$ of $t_{2g}$ electrons via intermediate Cl ions, (2) their direct NN hopping $t'$, (3) charge-transfer and cyclic exchange processes involving $pd$ charge-transfer excitations with energy $\Delta_{pd}$, and (4) the inter-orbital $t_{2g}$–$e_g$ hopping involving the strong $t_{pd\sigma}$ overlap between Cl-$p$ and Ru-$e_g$ orbitals. The calculations are standard but lengthy, and the details will be presented elsewhere.

Figure 6 shows the outcome of these calculations as a function of the trigonal crystal field $\Delta$, which controls the shape of the pseudospin wavefunction. In the cubic limit of $\Delta = 0$, Γ′ = 0 vanishes and $J$ is also very small, so that the NN exchange Hamiltonian is dominated by $K$ and Γ. For positive $\Delta$, the Heisenberg term $J$ is positive, and at large $\Delta$ it becomes comparable to the Kitaev interaction, while Γ decreases gradually. The observed FM $J < 0$ and the sizeable Γ value in RuCl₃ clearly point to a negative value of $\Delta$ in this compound. Indeed, from our analysis of the paramagnetic susceptibility (see Supplementary Note 4), we have obtained a negative $\Delta \simeq -50$ meV and ratio $\delta = 2\Delta/\lambda \simeq -0.7$. At this $\delta$ value, the calculated exchange constants are $(K, J, \Gamma, \Gamma') \simeq (-5, -2.2, 3.3, 1.3)$ meV (where we assumed the energy scale $t^2/U \simeq 12$ meV). The signs and relative values of these constants are quite consistent with the parameter set $(K, J, \Gamma, \Gamma') = (-5, -3, 2.5, 0.1)$ meV deduced from the RIXS experiment. Most importantly, the calculated Γ′ value is indeed the smallest among the NN exchange constants. This result is consistent with a recent ab-initio estimation of the exchange constants[60].

The overall behavior of the exchange parameters displayed in Fig. 6 is generic to $d^5$ Ru and Ir compounds. Indeed, using microscopic parameters appropriate for iridates[59,61], one obtains similar dependences $K(\delta)$, etc., with a sign change of $J$ near the cubic limit $\delta \sim 0$. Specifically, for Na₂IrO₃ with a positive $\delta \sim 0.75$[49], the calculations give an AFM $J \sim \Gamma \sim \frac{1}{2}|K|$ and a small Γ′ < 0, consistent with recent RIXS data[43]. The qualitative difference between RuCl₃ and Na₂IrO₃ can thus be primarily ascribed to the sign change of $\delta$, which leads to the sign change of $J$. The FM $J \sim \frac{1}{2}K$ with negative $\delta$ in RuCl₃ enhances the FM correlations and leads to the fragility of the zigzag order. On the other hand, the AFM $J \simeq \frac{1}{2}|K|$ in Na₂IrO₃ with positive $\delta$ leads to stable zigzag correlations up to 70 K[43]. These considerations highlight the trigonal field as an efficient control parameter of

magnetism in Kitaev materials. We also point out the analogy of $4d$ RuCl$_3$ and a new class of Kitaev materials based on $3d$ cobaltates[62,63]. In honeycomb cobaltates, both $K$ and $J$ are also of FM sign and the FM state is closely competing with the zigzag order[64]; consequently, the zigzag AFM order be suppressed at magnetic fields as small as ~1T[65].

It is worth noting here that our calculations assumed an ideal hexagonal symmetry, i.e. the exchange couplings $K$, $J$, etc. on the $x$, $y$, and $z$ type bonds are identical. While this is a reasonable approximation in the paramagnetic phase (where our experiment is performed), zigzag ordering below $T_N$ is expected to break this symmetry via pseudospin-lattice magnetoelastic coupling, resulting in exchange parameters on $z$-type bonds different from those on $x/y$ bonds along the zigzag direction (see Fig. 1). This coupling, which has been instrumental for understanding the low-energy magnon dynamics in the spin-orbit Mott insulator Sr$_2$IrO$_4$[66,67], should also be important in the AFM state of RuCl$_3$ and deserves future study.

To summarize, we have studied the Kitaev model material RuCl$_3$ by means of Ru $L_3$-edge RIXS. From the momentum dependence of the quasi-elastic RIXS intensity and its theoretical analysis, we have quantified the pseudospin-1/2 Hamiltonian parameters. The FM Kitaev term $K = -5$ meV is found to be the largest, but the Heisenberg exchange $J = -3$ meV and the off-diagonal exchange $\Gamma = 2.5$ meV are also significant. In particular, the large FM interaction $J$ is of crucial importance to explain the observed quick suppression of the short-ranged zigzag correlations above $T_N$. We found that the zigzag AFM order is only slightly lower in energy than the competing states, and the $q \sim 0$ correlations, typical for the FM Kitaev model and further enhanced by the large FM Heisenberg interaction $J \sim K/2$, become prominent as soon as the temperature is raised slightly above $T_N$. Our observation of the energetically proximate FM state also explains the quick suppression of the zigzag order by small magnetic fields. The precise nature of the metastable states governed by the highly frustrated Kitaev couplings remains an interesting open problem. The hierarchy of these states and the interplay between them may be influenced by sub-leading terms in the Hamiltonian, which could not be precisely determined by our measurements and analysis, and by higher-order or interlayer interactions we did not consider.

Based on the anisotropic $g$-factors $g_{ab} = -2.53$ and $g_c = -1.56$ determined from the Curie-Weiss analysis, we determined the trigonal field splitting $\Delta$ to be ~$-50$ meV (corresponding to trigonal elongation). The relatively weak trigonal splitting compared to the spin-orbit coupling $\lambda$ leads to an unquenched orbital moment and supports the notion of a spin–orbit entangled wavefunction. Using the microscopic parameters $10Dq = 2.4$, $J_H = 0.34$, and $\lambda = 0.15$ eV deduced from the high-energy multiplets of our RIXS spectra, we have evaluated the exchange constants $(K, J, \Gamma, \Gamma')$ also theoretically, reproducing the sign and hierarchy of the experimentally determined parameters. Notably, $\Gamma'$ is indeed at the bottom of the hierarchy for the experimentally relevant trigonal parameter $\delta$, and $J$ changes sign as a function of $\delta$. The FM $J$ obtained at negative $\delta$ in RuCl$_3$ is responsible for the fragility of the zigzag order, in contrast to its stability well above $T_N$ in Na$_2$IrO$_3$ with positive $\delta$ and AFM $J$. Overall, our findings form a solid basis for future theoretical and experimental studies of RuCl$_3$. In particular, the observed low-energy metastable states at $q \sim 0$, which are governed by the frustrated Kitaev and Heisenberg interactions, should be relevant for a quantitative understanding of the unusual field-induced properties of this material.

To determine the exchange Hamiltonian of RuCl$_3$, we have introduced a comprehensive approach that integrates information from RIXS data over an exceptionally wide range of energies and momenta. A two-pronged theoretical analysis of these spectra yields consistent results, inspiring confidence in the interaction parameters as a basis of future research on this material. In particular, systematic computation of equal-time correlation functions allowed us to evaluate quasi-elastic RIXS data as a fingerprint of the pseudospin interactions in real space. As RIXS requires only small crystals of characteristic dimensions ~10μm, our approach has the potential to evolve into a powerful screening tool for the rapidly expanding list of Kitaev candidate materials.

## Methods

**Sample growth and characterization**. RuCl$_3$ single crystals were grown by chemical vapor transport as reported previously[12]. Anhydrous polycrystalline RuCl$_3$ (99.9%, Acros Organics) was sealed in a ~12-cm-long quartz ampoule under vacuum. The reactant was heated at a rate of 3 K min$^{-1}$ to 1023 K for 120 h and then naturally cooled to room temperature. The reaction yielded shiny black crystalline platelets of RuCl$_3$ at the cooler end of the ampoule. The product was analyzed by powder x-ray diffraction and scanning electron microscopy, together with energy dispersive x-ray spectroscopy to check the purity of the crystals. The magnetic susceptibility of our crystals shows an anomaly at 7 K that corresponds to the Néel temperature ($T_N$) of the zigzag AFM order, but does not show any anomaly at 14 K[12] associated with an extrinsic magnetic transition caused by stacking faults[68,69].

**IRIXS spectrometer**. The RIXS spectra of RuCl$_3$ were collected using the newly-built intermediate x-ray energy RIXS spectrometer (IRIXS) at the Dynamics Beamline P01 of PETRA III, DESY[70–72]. The x-ray beam was focused to a beam spot of 20 μm × 150 μm (H × V). The horizontally scattered photons were collected at the scattering angle of 90° using a SiO$_2$ (10$\bar{2}$) diced spherical analyzer and a CCD camera, both placed in the Rowland geometry. The position of the zero-energy-loss line was determined by measuring non-resonant spectra from silver paint deposited next to the samples. The overall energy resolution of the IRIXS spectrometer at the Ru $L_3$-edge, defined as the FWHM of the non-resonant spectrum of silver, was ~100 meV. The variation of the x-ray beam footprint on the sample for different $\theta$ does not affect the total intensity in our detection scheme.

## Data availability

The data sets generated during and/or analyzed during the current study are available from the corresponding authors on reasonable request.

## Code availability

The numerical codes used to generate the results in this work are available from the corresponding authors on reasonable request.

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

## Acknowledgements

We are grateful to J.A. Sears and S. Francoual for sharing the magnetic REXS data prior to publication and Y.J. Kim and S.E. Nagler for enlightening discussions. We thank R.K. Kremer and E. Brücher for their assistance in the magnetization measurements. The project was supported by the European Research Council under Advanced Grant No. 669550 (Com4Com). We acknowledge DESY (Hamburg, Germany), a member of the Helmholtz Association HGF, for the provision of experimental facilities. The RIXS experiments were carried out at the beamline P01 of PETRA III at DESY. H.S. and K.U. acknowledge financial support from the JSPS Research Fellowship for Research Abroad.

H.S. is supported by the Alexander von Humboldt Foundation. J.C. acknowledges support by Czech Science Foundation (GAČR) under Project No. GA19-16937S. Computational resources were supplied by the project "e-Infrastruktura CZ" (e-INFRA LM2018140) provided within the program Projects of Large Research, Development, and Innovations Infrastructures.

## Author contributions

H.S., J.B., K.U., Z.Y., L.W., H.T., K.F., M.M., and H.G. performed the RIXS experiments. S.L., D.W., and B.V.L. grew $\alpha$-RuCl$_3$ single crystals and performed sample characterization. H.Y. and H.G. designed the beamline and IRIXS spectrometer. H.S. analyzed the experimental data. H.L., H.K., B.J.K., M.D., J.C., and G.K. carried out the theoretical calculations and contributed to the interpretation of the experimental data. H.S., H.L., G.K., and B.K. wrote the manuscript with contributions from all co-authors. B.K. initiated and supervised the project.

## Funding

## Competing interests

The authors declare no competing interests.
