## [Peer Review File · Nature Communications]

Reviewers' Comments:

Reviewer #1:

Remarks to the Author:

I am in strong support of accepting this manuscript.

I believe that the authors have reached an important conclusion by providing a careful analysis of their rather comprehensive set of the resonant inelastic X-ray scattering data on an important material of great current interest. As such, their study can be expected to impact research in the area of quantum materials and novel quantum states, broadly defined, and that of the Kitaev materials specifically.

However, I would like to ask the authors to modify some of their assertions in order to reflect more carefully on the mechanisms they allude to and to provide a better interpretation to their own achievements. Please, see my detailed comments below.

In brief, the present study uses resonant inelastic X-ray scattering (RIXS) to determine some of the key microscopic parameters of aRuCl_3 , such as crystal field, spin-orbit, and Hund's couplings. These are subsequently used to theoretically limit parameters of the low-energy pseudo-spin model (generalized Kitaev model, or $\text{KJGG}'\text{-J3}$ model) using a perturbative downfolding approach. On top of that, the same pseudo-spin model is analyzed for some selected set of parameters in order to fit some of the RIXS results directly using exact diagonalization on finite-size clusters. From this analysis a conclusion is drawn of aRuCl_3 being in a proximity to a ferromagnetic (FM) state, with predominant FM correlations observed in the paramagnetic state, while the observed zigzag order is stabilized by strong quantum fluctuations.

I fully agree with the main message of the work that aRuCl_3 is proximate to a (fluctuating) FM state.

My strongest objection is, at least partially, terminological.

The authors invoke quantum order-by-disorder (ObD) mechanism (Abstract and lines 241-249) as being responsible for selection between competing states. First, the hallmark of the ObD is the presence of an **extensive**, i.e., macroscopic, degeneracy of the ground state (typically on a classical level), which is being lifted by quantum effects. Such a degeneracy **has not been** demonstrated in this work for the advocated choices of parameters of the pseudo-spin model. In fact, I am not aware of such an analysis for the KJGG' model except for the K-only and some special K-J model limits. Second, The ObD is a very delicate phenomenon, with the associated energy gain typically in very small fraction of the typical exchange constant. The authors seem to argue for a rather monstrous energy shift by $\sim 1\text{meV}$ (Fig. 1, some $|K|/5$). Lastly, even in the most well-characterized and well-argued ObD cases (pyrochlores) it is often very hard to actually prove that the order selection is not due to some subleading interactions. In the present study, the authors just use a sweeping assertion that the ObD must be the case, without such an analysis.

In my opinion, proximity to the FM state is a strong enough statement.

Why undermine it by a questionable interpretation?

Quantum fluctuations **can** shift and are known to shift phase boundaries. Why not just argue for strong quantum corrections to (fluctuations in) the ground state?

My second criticism concerns claimed model parameter set. The rather sweeping conclusion on the invalidity of the classical treatment of the model seems to be made rather hastily based on the comparison of only a handful of representative sets and without a systematic analysis. Compared with the prior choices, the main change seems to be in a larger (negative, or FM) J . In their downfolding analysis, the authors also find non-negligible and positive G' -term, which is in agreement with some recent analysis, but they do not use it in their RIXS fits. Why?

To insist that somehow their choice of parameters is an ultimate one for further studies is really an overstatement. The downfolding method is perturbative and involves some ad hoc choice of the hopping parameters, and the ab-initio methods the authors compare with yields many terms that are necessarily truncated.

I, thus, suggest to emphasize positive achievements and to downplay vehemence of some of the claims regarding parameter choice.

A couple of small comments.

— Lines 139-141: “Interestingly, this energy scale is roughly consistent with the Zeeman energy of $S = 1/2$ under a magnetic field of ~ 8 T, where the zigzag order disappears”
The scale of 8T magnetic field is *the same* as that of TN and, arguably *is* just that scale. I do not seem to appreciate the word “Interestingly” here.

— In introduction, lines 52-54, a reference is made to “fermionic temperature dependence over a wide temperature range was indeed revealed by Raman scattering”. It is worth noting that the Brillouin function of paramagnetic $S=1/2$ at high temperatures is purely fermionic. The trend sets already for $T > J/2$ even in the unfrustrated models. The observed “fermionicity” was made for $T > 100$ K, which is well above $|K|/2 = 25$ K.

Reviewer #2:

Remarks to the Author:

The manuscript by H. Suzuki, et. al. reported the Ru L3 edge resonant inelastic x-ray scattering measurements and analyses of α -RuCl₃. High energy orbital excitations and multiplet calculations provide accurate estimations of important local interaction parameters such as crystal-field, Hund's, and spin-orbit coupling. The main message in this manuscript is drawn from the momentum-transfer dependence of the intensity modulation in the low energy excitation near the zero-energy loss. They found that the existing sets of the pseudospin Hamiltonian parameter do not reproduce their experimental data. New set of parameter values drawn in this manuscript suggest enhanced ferromagnetic correlations whose implications are discussed.

I totally agree that the high statistics of the orbital excitations in this work provide accurate estimations of important local interaction parameters such as crystal-field, Hund's, and spin-orbit coupling which are, in my opinion, not possible in other spectroscopy techniques such as inelastic x-ray scattering.

The majority of low energy magnetic excitations in α -RuCl₃ exist inside 10 meV (from the inelastic neutron scattering studies) but the highest achievable energy resolution of the Ru L3 edge RIXS is about 100 meV at the moment. Therefore, they cannot help but use the methodology of the intensity modulation analysis for the main result. Peak energy estimation may be possible even with the one order of magnitude poor energy resolution. But I am not quite sure whether the one order of magnitude poor energy resolution can be used for an accurate parameter estimation from dispersing excitations. Nonetheless, my criticism is on whether the data were treated properly, and the intensity modulation data and analysis advance the current understanding of the physics in α -RuCl₃ which has been obtained from the high-resolution inelastic neutron scattering studies.

The photon energy at the Ru L3 edge is high enough so that the whole Brillouin zone can be accessed. But they still have to use very large sample angle range which drastically affects the excitation intensity modulation. They carried out the self-absorption corrections based on the work of Tröger, L. et al. Actually, this self-absorption correction is about correcting the intensity vs. energy shape by considering the x-ray attenuations with a finite solid angle detector. In my knowledge, it is about the point source. How about the intensity change due to the different size of the illuminated area? The footprint is about 860 μ m at the grazing incident of 10 deg but 152 μ m at the normal incident of 80 deg. The soft x-ray RIXS faces the same intensity change problem. The soft x-ray RIXS data are corrected by matching high energy orbital excitation intensity and then carrying out the self-absorption correction. But, in my knowledge, it works for some range of spectra but not the whole range of spectra and there is no absolute solution for it. In this regard, the intensity analysis has not been used for accurate estimations of spectroscopic information. This work does not do the footprint correction.

The main result is that the low energy spectral weight is highest at the zone center. Enhanced ferromagnetic correlations explain such spectral weight distribution. Other sets of parameters from Ref. 20 and 38 predict that the low energy spectral weight is highest other than the zone center. Here a simple question has come to my mind: does the parameter set in this work describes the energy spectra from the inelastic neutron scattering measurements?. The Ref. 20 parameter set

differ only in the J value from this work: $J=-0.5$ in Ref.20 and $J=-2.5$ in this work. It seems clear that the parameter set in this work does not correctly describe the energy spectra from the inelastic neutron scattering measurements, considering that the Ref. 20 parameter set nicely describes it. The suggested enhanced ferromagnetic correlations in this work will appear as a soft mode at the zone center whose large spectral weight energy sits mostly below 2meV. The dispersion relation will be very different from Ref. 20 and 38: soft modes at the zone center and zone boundary.

I think authors should simulate the RIXS intensity outside the quasi-elastic range below 2meV to see the dynamic components of the pseudospin response. Inelastic neutron scattering measurements clearly show that the largest spectral weight of finite-energy magnetic excitations exists at the zone center at all temperatures. Here, the problem is that one order of magnitude poor energy resolution of the Ru L3 edge resonant inelastic x-ray scattering could not distinguish finite-energy magnetic excitations from the quasi-elastic signals below 2meV.

The data treatment and the analysis are not flawless. I think authors do their best to address the issues in α -RuCl₃ but could not overcome the fundamental technique limitation of the one order of magnitude energy resolution mismatch. The manuscript by H. Suzuki, et. al. is not recommended for the publication.

Reply to the Referees for the manuscript NCOMMS-20-47832

Correction to Table I

The RIXS intensity calculation for Ref. [38] in the former version was not for the model 1 of Ref. [38] but for model 2: $(K, J, \Gamma, \Gamma', J_3) = (-10, -1.5, 8.8, 0, 0.4)$. Additionally, the classical energy of the ferromagnetic state ($E_{\text{FM}} - E_{\text{ZZ}}$) was misquoted by a factor of $1/S^2=4$. The correct information is included in the revised manuscript.

Reply to Reviewer #1

Comment:

I am in strong support of accepting this manuscript.

I believe that the authors have reached an important conclusion by providing a careful analysis of their rather comprehensive set of the resonant inelastic X-ray scattering data on an important material of great current interest. As such, their study can be expected to impact research in the area of quantum materials and novel quantum states, broadly defined, and that of the Kitaev materials specifically. However, I would like to ask the authors to modify some of their assertions in order to reflect more carefully on the mechanisms they allude to and to provide a better interpretation to their own achievements. Please, see my detailed comments below.

In brief, the present study uses resonant inelastic X-ray scattering (RIXS) to determine some of the key microscopic parameters of aRuCl₃, such as crystal field, spin-orbit, and Hund's couplings. These are subsequently used to theoretically limit parameters of the low-energy pseudo-spin model (generalized Kitaev model, or KJGG'-J₃ model) using a perturbative downfolding approach. On top of that, the same pseudo-spin model is analyzed for some selected set of parameters in order to fit some of the RIXS results directly using exact diagonalization on finite-size clusters. From this analysis a conclusion is drawn of aRuCl₃ being in a proximity to a ferromagnetic (FM) state, with predominant FM correlations observed in the paramagnetic state, while the observed zigzag order is stabilized by strong quantum fluctuations.

Our Reply:

We are grateful to Reviewer #1 for his/her strong recommendation for the acceptance of our manuscript and for appreciating its broad impact on the research area of quantum materials. In accordance with the Reviewer's comments, we have revised our assertions as detailed below.

Comment:

I fully agree with the main message of the work that aRuCl₃ is proximate to a (fluctuating) FM state.

Our Reply:

We thank Reviewer #1 for agreeing on the main conclusion of our work.

Comment:

My strongest objection is, at least partially, terminological.

The authors invoke quantum order-by-disorder (ObD) mechanism (Abstract and lines 241-249) as being responsible for selection between competing states. First, the hallmark of the ObD is the presence of an *extensive*, i.e., macroscopic, degeneracy of the ground state (typically on a classical level), which is being lifted by quantum effects. Such a degeneracy *has not been* demonstrated in this work for the advocated choices of parameters of the pseudo-spin model. In fact, I am not aware of such an analysis for the KJGG' model except for the K-only and some special K-J model limits. Second, The ObD is a very delicate phenomenon, with the associated energy gain typically in very small fraction of the typical exchange constant. The authors seem to argue for a rather monstrous energy shift by $\sim 1\text{meV}$ (Fig. 1, some $|K|/5$). Lastly, even in the most well-characterized and well-argued ObD cases (pyrochlores) it is often very hard to actually prove that the order selection is not due to some subleading interactions. In the present study, the authors just use a sweeping assertion that the ObD must be the case, without such an analysis.

In my opinion, proximity to the FM state is a strong enough statement. Why undermine it by a questionable interpretation? Quantum fluctuations *can* shift and are known to shift phase boundaries. Why not just argue for strong quantum corrections to (fluctuations in) the ground state?

Our Reply:

We highly appreciate the Reviewer's helpful and constructive critique. We agree that it is more appropriate to refer to "quantum fluctuations" here, rather than the "order-by-disorder" mechanism, which is often understood in the narrower sense described by the Reviewer. We have revised the corresponding text as follows.

Abstract:

We have removed the term "order-by-disorder". The revised sentence reads "The zigzag state is stabilized by quantum fluctuations, leaving ferromagnetism – along with the Kitaev spin liquid – as energetically proximate metastable states."

Line 244:

We have removed the phrase "in the spirit of the order-from-disorder physics,".

Comment:

My second criticism concerns claimed model parameter set. The rather sweeping conclusion on the invalidity of the classical treatment of the model seems to be made rather hastily based on the comparison of only a handful of representative sets and without a systematic analysis. Compared with the prior choices, the main change seems to be in a larger (negative, or FM) J. In

their downfolding analysis, the authors also find non-negligible and positive G' -term, which is in agreement with some recent analysis, but they do not use it in their RIXS fits. Why?

Our Reply:

We are grateful to Reviewer #1 for the critical comment that has guided us to improve the pseudospin model. As stated in the original manuscript, we assumed $\Gamma' = 0$ to reduce the free parameter space in the RIXS fit based on the absence of clear splitting of the $S = 3/2$ transitions. However, as the Reviewer pointed out, the inclusion of a non-vanishing Γ' term is required to coherently account for the momentum dependence of RIXS intensity and the nonzero trigonal field inferred from the anisotropic magnetic susceptibility.

In the revised manuscript, we have further refined the pseudospin model by allowing a nonzero Γ' . The revised parameter set reads $(K, J, \Gamma, \Gamma', J_3) = (-5, -3, 2.5, 0.1, 0.75)$, which – in addition to the nonzero Γ' – also includes a slightly enhanced J and J_3 as compared with the previous set (termed “Alternative 1” in the revised manuscript). Note here that the Γ' term is indeed positive and at the bottom of the hierarchy. This revised set not only excellently reproduces the momentum dependence of the RIXS intensity (Figs. 5a and b), but also captures the pseudospin dynamics revealed by inelastic neutron scattering (INS). In response to Reviewer #2’s comment, we have calculated the pseudospin dynamical structure factor. The trace of the pseudospin susceptibility for different parameter sets is now included in the supplementary material as Fig. S3. The comparison reveals that the revised set better captures the INS peak around 2 meV reported in Ref. [55].

In accordance with the above revisions, the parameter values in the main text have been revised. Furthermore, we have added a new section “Supplementary Note 3: Selection of the pseudospin Hamiltonian parameters” in the supplementary Information. This new section addresses how a slight deviation of interaction parameters from the optimal parameter set results in differences in the simulated RIXS intensity and pseudospin dynamics. Comparisons among the optimal set, the Alternatives 1 and 2, and the model of Winter et al. (the starting point of our RIXS fit) will convey the reader how we have reached the optimal set.

Comment:

To insist that somehow their choice of parameters is an ultimate one for further studies is really an overstatement. The downfolding method is perturbative and involves some ad hoc choice of the hopping parameters, and the ab-initio methods the authors compare with yields many terms that are necessarily truncated. I, thus, suggest to emphasize positive achievements and to downplay vehemence of some of the claims regarding parameter choice.

Our Reply:

We certainly do not wish to claim that the parameters we are quoting are the final word on this issue. We thank the Reviewer for pointing out the misleading impression created by some of the wording we had used in the original version of our manuscript. We have critically examined

the manuscript and modified the text as follows to avoid such an impression. If the Reviewer finds other statements that could be read in this way, we will be glad to modify these as well.

Line 62 in the Introduction: We inserted the qualifier "... the leading terms in ..."

"In the present work, we determined *the leading terms in* the pseudospin Hamiltonian of RuCl_3 ..."

Line 325 in Conclusions: We inserted the following sentence.

"The hierarchy of these states and the interplay between them may be influenced by sub-leading terms in the Hamiltonian, which could not be precisely determined by our measurements and analysis, and by higher-order or interlayer interactions we did not consider."

Comment:

— Lines 139-141: "Interestingly, this energy scale is roughly consistent with the Zeeman energy of $S=1/2$ under a magnetic field of ~ 8 T, where the zigzag order disappears"

The scale of 8T magnetic field is *the same* as that of TN and, arguably *is* just that scale. I do not seem to appreciate the word "Interestingly" here.

Our Reply:

Following the Reviewer's suggestion, we have removed the word "Interestingly" in the revised manuscript.

Comment:

In introduction, lines 52-54, a reference is made to "fermionic temperature dependence over a wide temperature range was indeed revealed by Raman scattering". It is worth noting that the Brillouin function of paramagnetic $S=1/2$ at high temperatures is purely fermionic. The trend sets already for $T>J/2$ even in the unfrustrated models. The observed "fermionicity" was made for $T>100\text{K}$, which is well above $|K|/2=25\text{K}$.

Our Reply:

We appreciate the Reviewer for pointing out this mistake. We have removed this statement from the introduction as it is not of particular importance in the context. The revised sentence reads "In RuCl_3 , a magnetic scattering continuum has been observed by Raman scattering [25] and inelastic neutron scattering experiments [26–29].".

Reply to Reviewer #2

Comment:

The manuscript by H. Suzuki, et. al. reported the Ru L3 edge resonant inelastic x-ray scattering measurements and analyses of $\alpha\text{-RuCl}_3$. High energy orbital excitations and multiplet calculations provide accurate estimations of important local interaction parameters such as crystal-field, Hund's, and spin-orbit coupling. The main message in this manuscript is drawn from the momentum-transfer dependence of the intensity modulation in the low energy

excitation near the zero-energy loss. They found that the existing sets of the pseudospin Hamiltonian parameter do not reproduce their experimental data. New set of parameter values drawn in this manuscript suggest enhanced ferromagnetic correlations whose implications are discussed.

I totally agree that the high statistics of the orbital excitations in this work provide accurate estimations of important local interaction parameters such as crystal-field, Hund's, and spin-orbit coupling which are, in my opinion, not possible in other spectroscopy techniques such as inelastic x-ray scattering.

Our Reply:

We thank Reviewer #2 for appreciating the high quality of our RIXS data and their unique capability of determining the local interaction parameters and the pseudospin Hamiltonian.

Comment:

The majority of low energy magnetic excitations in α -RuCl₃ exist inside 10 meV (from the inelastic neutron scattering studies) but the highest achievable energy resolution of the Ru L₃ edge RIXS is about 100 meV at the moment. Therefore, they cannot help but use the methodology of the intensity modulation analysis for the main result. Peak energy estimation may be possible even with the one order of magnitude poor energy resolution. But I am not quite sure whether the one order of magnitude poor energy resolution can be used for an accurate parameter estimation from dispersing excitations. Nonetheless, my criticism is on whether the data were treated properly, and the intensity modulation data and analysis advance the current understanding of the physics in α -RuCl₃ which has been obtained from the high-resolution inelastic neutron scattering studies.

Our Reply:

As the Reviewer pointed out, the intrinsic bandwidth of spin excitations in α -RuCl₃ is of the order of 10 meV, while the energy resolution at the Ru L₃ edge is 100 meV. This is the reason why we do not discuss the energy dispersion of the $S = 1/2$ excitations. Nevertheless, the total spectral weight and its momentum distribution can be reliably determined, since the $S = 3/2$ transitions are well separated from the quasi-elastic peaks, and the extrinsic contribution to the elastic peak is significantly smaller than the intrinsic magnetic scattering (see Supplementary Information). We would like to emphasize that the \mathbf{q} dependence of integrated RIXS intensity, in conjunction with theoretical modelling, narrowly constrains the exchange interaction parameters of the extended Kitaev-Heisenberg Hamiltonian. We will show below that our results and conclusions are completely consistent with published inelastic neutron scattering results. We also stress that our RIXS approach is universally applicable to Kitaev candidate materials, many of which are not amenable to inelastic neutron scattering due to the absence of large single crystals.

Comment:

The photon energy at the Ru L₃ edge is high enough so that the whole Brillouin zone can be accessed. But they still have to use very large sample angle range which drastically affects the

excitation intensity modulation. They carried out the self-absorption corrections based on the work of Tröger, L. et al. Actually, this self-absorption correction is about correcting the intensity vs. energy shape by considering the x-ray attenuations with a finite solid angle detector. In my knowledge, it is about the point source. How about the intensity change due to the different size of the illuminated area? The footprint is about 860 μm at the grazing incident of 10 deg but 152 μm at the normal incident of 80 deg. The soft x-ray RIXS faces the same intensity change problem. The soft x-ray RIXS data are corrected by matching high energy orbital excitation intensity and then carrying out the self-absorption correction. But, in my knowledge, it works for some range of spectra but not the whole range of spectra and there is no absolute solution for it. In this regard, the intensity analysis has not been used for accurate estimations of spectroscopic information. This work does not do the footprint correction.

Our Reply:

We would like to clarify that the self-absorption effect in the current context is the absorption of x-rays along the scattering path before and after the scattering event, and is dependent solely on the geometrical configurations of the sample, the incoming beam, and the scattered beam. It is therefore independent of the actual x-ray detection scheme. As the reviewer pointed out, Eq. (2) in the work of Tröger *et al.* takes into account not only the geometrical configurations but also the effect of attenuation in their specific detector. To avoid any confusion, we have changed the reference to M. Minola *et al.*, PRL 114, 217003 (2015) in the revised manuscript.

We now further consider the effect of the evolution of the beam footprint. The beam profile of the incident beam is of 20 μm \times 150 μm (H \times V), and the x-ray photons are scattered horizontally. Along the $\mathbf{q} = (H, 0)$ path, the illuminated area develops from 164 μm \times 150 μm ($H = -1$) to 20.2 μm \times 150 μm ($H = 1$). In our detection scheme detailed in Ref. [72], the scattered photons are dispersed by the diced spherical analyzers and focused onto the pixels of the CCD detector depending on the x-ray energy. The broadening of the beam footprint along the horizontal direction causes a broadening of the focal point on the CCD along the horizontal direction. However, to obtain the RIXS spectra, we sum up the CCD intensity along the horizontal direction. Therefore, the total intensity is perfectly conserved even in the presence of footprint broadening. Moreover, the variation in the beam propagation angle at the diced analyzers, located at 100 mm away from the sample, is at most $\sim 164 \mu\text{m} / 100 \text{ mm} \sim 1.64 \text{ mrad} = 0.09 \text{ degrees}$. This variation is much smaller than the sample angle ranges (7-83 degrees) used to map out the dispersion, and we did not observe any strong intensity modulation within such a small angle range.

To convey the above considerations, we have revised the corresponding text as follows (Methods, lines 364 and 369).

Methods, IRIXS spectrometer:

“The x-ray beam was focused to a beam spot of 20 μm \times 150 μm (H \times V). The horizontally scattered photons were...”

“The variation of the x-ray beam footprint on the sample for different θ does not affect the total intensity in our detection scheme.”

Comment:

The main result is that the low energy spectral weight is highest at the zone center. Enhanced ferromagnetic correlations explain such spectral weight distribution. Other sets of parameters from Ref. 20 and 38 predict that the low energy spectral weight is highest other than the zone center. Here a simple question has come to my mind: does the parameter set in this work describes the energy spectra from the inelastic neutron scattering measurements?. The Ref. 20 parameter set differ only in the J value from this work: $J=-0.5$ in Ref.20 and $J=-2.5$ in this work. It seems clear that the parameter set in this work does not correctly describe the energy spectra from the inelastic neutron scattering measurements, considering that the Ref. 20 parameter set nicely describes it. The suggested enhanced ferromagnetic correlations in this work will appear as a soft mode at the zone center whose large spectral weight energy sits mostly below 2meV. The dispersion relation will be very different from Ref. 20 and 38: soft modes at the zone center and zone boundary.

I think authors should simulate the RIXS intensity outside the quasi-elastic range below 2meV to see the dynamic components of the pseudospin response. Inelastic neutron scattering measurements clearly show that the largest spectral weight of finite-energy magnetic excitations exists at the zone center at all temperatures. Here, the problem is that one order of magnitude poor energy resolution of the Ru L3 edge resonant inelastic x-ray scattering could not distinguish finite-energy magnetic excitations from the quasi-elastic signals below 2meV.

Our Reply:

We are grateful to Reviewer #2 for this excellent suggestion. We do agree that the published inelastic neutron scattering (INS) data allow an additional cross-check of the pseudospin Hamiltonian extracted from RIXS. We note, however, that the latest set of high-resolution INS data (Balz *et al.*, Ref. [55]) exhibit sharper features than the earlier data by Banerjee *et al.* (Ref. [27]), and less intense continuum than reported in the early data. This evolution is not unusual, as the sample quality improves and the experimental conditions are optimized when research progresses.

Following the Reviewer’s suggestion, we have developed a new set of numerical routines to compute the pseudospin dynamical structure factor for the different pseudospin models. These results are described in a new Section of the Supplementary Materials. The central results are displayed in Fig. S3, which shows that the trace of the pseudospin susceptibility for the optimal set precisely captures the well-defined single peak around 2 meV at the Γ point seen in the highest-quality set of INS data [55]. On the other hand, the parameters from Winter *et al.* [20], which were developed to describe the earlier INS data with broader features, fail to capture the pronounced peak at the Γ point (Fig. S3d).

Comment:

The data treatment and the analysis are not flawless. I think authors do their best to address the issues in α -RuCl₃ but could not overcome the fundamental technique limitation of the one order of magnitude energy resolution mismatch. The manuscript by H. Suzuki, et. al. is not recommended for the publication.

Our Reply:

The reassessment of the pseudospin Hamiltonian based on the pseudospin dynamical response has shown excellent agreement both with our RIXS data and with the best INS data that are currently available. We are thus confident that the revised model will serve as a solid foundation for further studies on RuCl₃.

Reviewers' Comments:

Reviewer #1:

Remarks to the Author:

I have read the new version of the manuscript and the authors' responses.

The level of the revisions, their depth and clarity, are really admirable.

The authors have provided a comprehensive response to all my comments, and I believe also addressed mostly technical concerns of the other referee.

The paper is very well written, clear, and convincing.

It will be an influential publication.

I accept it without hesitation.

Reviewer #2:

Remarks to the Author:

In the revised manuscript, authors addressed all comments. The manuscript by H. Suzuki, et. al. is recommended for the publication.

Reply to the Referees for the manuscript NCOMMS-20-47832A

Reply to Reviewer #1

Comment:

I have read the new version of the manuscript and the authors' responses. The level of the revisions, their depth and clarity, are really admirable. The authors have provided a comprehensive response to all my comments, and I believe also addressed mostly technical concerns of the other referee. The paper is very well written, clear, and convincing. It will be an influential publication. I accept it without hesitation.

Our Reply:

We are grateful to Reviewer #1 for highly appreciating our revised manuscript and his/her recommendation for publication.

Reply to Reviewer #2

Comment:

In the revised manuscript, authors addressed all comments. The manuscript by H. Suzuki, et. al. is recommended for the publication.

Our Reply:

We thank Reviewer #2 for the recommendation for the publication.